# Sublinear Time Algorithms for Greedy Selection in High Dimensions

**Qi Chen**[*1]          **Kai Liu**[*2]          **Ruilong Yao**[2]          **Hu Ding**[†2]

[1]School of Data Science, University of Science and Technology of China, Anhui, China
[2]School of Computer Science and Technology, University of Science and Technology of China, Anhui, China

## Abstract

Greedy selection is a widely used idea for solving many machine learning problems. But greedy selection algorithms often have high complexities and thus may be prohibitive for large-scale data. In this paper, we consider two fundamental optimization problems in machine learning: $k$-center clustering and convex hull approximation, where they both can be solved via greedy selection. We propose sublinear time algorithms for them through combining the strategies of randomization and greedy selection. Our results are similar in spirit to the linear time stochastic greedy selection algorithms for submodular maximization [Mirzasoleiman et al., AAAI 2015, Hassidim and Singer, ICML 2017], but with several important differences. Our runtimes are independent of the number of input data items $n$. In particular, our runtime for $k$-center clustering significantly improves upon that of the uniform sampling approach, especially when the dimensionality is high. Our sublinear algorithms can also reduce the computational complexities for various applications, such as data selection and compression, active learning, and topic modeling, *etc*.

## 1 INTRODUCTION

Greedy algorithm is one of the most fundamental tools for algorithm design [Cormen et al., 2009]. In particular, many optimization problems in machine learning can be solved through *greedy selection* method. The method iteratively selects a subset of data items from input based on some greedy strategy. One representative example is the Gonzalez's algorithm for *k-center clustering* [Gonzalez, 1985]. Given a

set of data items (*e.g.,* a point set in $\mathbb{R}^d$), the algorithm is to iteratively select $k$ items from the input; if one draws $k$ equal-sized balls centered at these $k$ items, the whole input data set can be covered by these balls and the radius is no larger than two times the optimal one (the formal definition for $k$-center clustering is shown in Section 2).

The algorithm is simple but has many important applications in real world. One natural application is constructing *coreset* for compressing a large-scale data, especially when we want to maximize diversity or coverage [Indyk et al., 2014]. Another closely related application is *batch active learning* [Sener and Savarese, 2018, Coleman et al., 2020]. Most machine learning models heavily depend on high-quality labeled training datasets. However, because it is expensive to acquire a large number of labeled data, we may only be able to select a small number of data items (via $k$-center clustering) to label in each round (as an active learning process).

Another high dimensional optimization problem that depends on greedy selection is *convex hull approximation* [Blum et al., 2019, Awasthi et al., 2020], where the goal is to find a convex hull so that each data item can be approximately represented by the vertices. The problem has a number of applications in machine learning, such as topic modeling, sparse approximation, and non-negative matrix factorization [Ge and Moitra, 2020]. Though the convex hull algorithms have been well studied in low dimensions [de Berg et al., 2008], the problem in high dimensions is much more challenging. Similar with the $k$-center clustering, a common idea for convex hull approximation is using greedy selection to find the vertices iteratively.

Although these greedy selection methods enjoy promising performances in practice, they often suffer from high complexities when data sizes are extremely large. For instance, the vanilla Gonzalez's algorithm needs to run $k$ iterations and each iteration needs to scan the whole dataset in one pass (a detailed introduction on the previous work is shown in Section 1.1). Similarly, the greedy selection method for

---

[*]The first two authors contributed equally to this work.
[†]Corresponding author.

*Accepted for the 38[th] Conference on Uncertainty in Artificial Intelligence* (UAI 2022).

convex hull approximation also needs to repeatedly scan the whole dataset and thus yields large runtime. So a natural question is:

*Can we modify these greedy selection algorithms to achieve lower time complexities,* e.g., *sublinear time complexities that are independent of input data size, and meanwhile preserve their quality guarantees?*

## 1.1 RELATED WORK

We introduce several important existing results related to this paper in this section.

$k$-**center clustering.** As mentioned before, the greedy selection based $k$-center clustering algorithm [Gonzalez, 1985] can yield a 2-approximation result; moreover, it was proved that any approximation ratio lower than 2 implies $P = NP$ [Hochbaum and Shmoys, 1985]. To speed up the Gonzalez's algorithm, several improvements have been proposed before [Feder and Greene, 1988, Har-Peled and Mendel, 2006]; however, they usually require some additional assumptions (*e.g.,* the dimensionality or intrinsic dimensionality should be small). To deal with large-scale data, a number of streaming algorithms which only need to read the data in one-pass were introduced in Charikar et al. [2004], McCutchen and Khuller [2008], Guha [2009], Ceccarello et al. [2019]. The well known "coreset" technique is also applied to compress data size for $k$-center clustering Bādoiu et al. [2002], Aghamolaei and Ghodsi [2019], but their coreset construction algorithms already take at least linear time. Furthermore, several uniform sampling based ideas were presented for achieving sublinear complexity for $k$-center clustering (with outliers) [Charikar et al., 2003, Huang et al., 2018].

**Convex hull approximation.** Several elegant convex hull algorithms for low-dimensional space have been introduced in the community of computational geometry before [de Berg et al., 2008]. The high-dimensional convex hull approximation problem is closely related to non-negative matrix factorization and topic modeling [Ge and Moitra, 2020]. Roughly speaking, the vertices of the obtained convex hull can help us to generate the low rank non-negative matrices and discover the hidden topics. In general, this problem is intractable but it is possible to achieve an efficient solution under the *separability* assumption [Donoho and Stodden, 2003, Arora et al., 2012]. Recently, several practical algorithms with provable guarantees were also proposed, such as Blum et al. [2019], Awasthi et al. [2020], Arora et al. [2013].

**Other applications of greedy selection in machine learning.** Besides the aforementioned two problems, greedy selection also has several other applications in machine learning. To name a few: *submodular maximization* [Nemhauser et al., 1978], *column subset selection* [Farahat et al., 2015], *reinforcement learning* [Painter-Wakefield and Parr, 2012], *sparse approximation* [Tropp, 2004], and SVM Gärtner and Jaggi [2009].

## 1.2 OUR CONTRIBUTIONS

In this paper, we aim to develop sublinear time algorithms for the $k$-center clustering and convex hull approximation problems. We assume the input data size and the dimensionality are both large. We combine the strategies of greedy selection and randomization, and show that the randomized greedy selection methods can achieve almost the same approximation guarantees, and meanwhile, the time complexities can be reduced to be sublinear.

**Comparison with the algorithms of Mirzasoleiman et al. [2015], Hassidim and Singer [2017].** Actually, the high complexity issue of greedy selection has been discussed in Mirzasoleiman et al. [2015], Hassidim and Singer [2017] for the submodular maximization problem. They showed that if the greedy selection step is replaced by random sampling, a quality guarantee still holds but the complexity (*i.e.,* the number of function evaluations) can be reduced to be linear. Our proposed algorithms are inspired by the similar stochastic intuition but with several important differences. **(i)** First, both $k$-center clustering and convex hull approximation are geometric optimizations in high dimensions which have different objective functions other than submodular maximization. **(ii)** Second, our framework yields the sublinear time complexities that are independent of the number of input data items; this property is particularly important when we cannot access the input data and can only take a small sample via an oracle each time (*e.g.,* due to privacy preserving or the challenge of data acquisition). **(iii)** Finally, we also consider the scenario that the number of iterations for greedy selection is unknown. For example, the number of clusters "$k$" of the $k$-center clustering may not be given; instead, we may just run the Gonzalez's algorithm iteratively until the obtained radius is no larger than a pre-specified threshold $r_0 > 0$. We need to emphasize that designing the sublinear time algorithm becomes much more challenging with such a change, since it will be difficult to set the sample size in each iteration and determine when the algorithm should terminate. To remedy these issues, we propose a novel stratified sampling method and design a sampling based stopping condition for the greedy selection.

**Comparison with the streaming and uniform sampling algorithms.** As mentioned in Section 1.1, the one-pass streaming algorithms [Charikar et al., 2004, McCutchen and Khuller, 2008, Guha, 2009, Ceccarello et al., 2019] can avoid repeatedly reading the input data, however, they still suffer from high time complexities (*e.g.,* the "doubling algorithm" [Charikar et al., 2004] takes a total $O\big(k(\log k)nd\big)$ time that is even higher than the complexity of the vanilla Gonzalez's algorithm, where $n$ is the number of input

points). On the other hand, our proposed sublinear time algorithms have the complexities independent of $n$.

It is also worth to compare our results with the uniform sampling algorithms for $k$-center clustering [Charikar et al., 2003, Huang et al., 2018]. For example, a simple uniform sample $S$ of size $\tilde{O}(\frac{kd}{\epsilon^2})$ [1] can approximately represent the whole input data $P$ based on the theory of VC dimension [Huang et al., 2018], where $\epsilon \in (0, 1)$ indicates the small fraction of uncovered points; that is, if one runs the 2-approximate Gonzalez's algorithm on the sample $S$, the obtained $k$ balls still form a 2-approximate solution in terms of the whole input $P$ but except for $\epsilon n$ uncovered points of $P$. The running time of the Gonzalez's algorithm on $S$ should be $\tilde{O}(k|S|d) = \tilde{O}(\frac{k^2d^2}{\epsilon^2})$. In Section 3.1, we show that our algorithm takes $\tilde{O}(\frac{k^3d}{\epsilon})$ time (also with $\epsilon n$ uncovered points). Usually, $k$ is much smaller than the dimensionality $d$, and thus our improvement is significant. In particular, if $k$ is assumed to be constant, we improve their complexity by a factor up to $\frac{d}{\epsilon}$.

The reader may wonder that whether dimension reduction technique (*e.g.*, the JL-transform Dasgupta and Gupta [2003]) can be applied. Actually the complexities of both our method and the uniform sampling can be reduced by the JL-transform, and our improvement is still significant (just replace the dimension $d$ by the new dimension $d'$ for both the two complexities). Also, even we apply the JL-transform, the reduced dimensionality could be still high (which is $O(\log|S|/\mu^2)$, if supposing $|S|$ is the total sample size and "$\mu$" is the pairwise distance distortion error). For example, if we let $\mu = 0.01$, the new dimension is still high.

Not only the runtime, another benefit comparing with the uniform sampling is that we have smaller sample size. Our algorithm takes $\tilde{O}(k^2/\epsilon)$ samples in total after $k$ iterations, which is much lower than $\tilde{O}(kd/\epsilon^2)$ if assuming $k$ is not large. In particular, **our sample size is independent of the dimension** $d$. The dimension can be very high or even infinity if using kernel. The smaller sample size is also important in some specific setting like relational database [Schleich et al., 2019]. It is very expensive to materialize the whole data matrix for a relational database, and a smaller sample size can significantly reduce the total computational complexity [Zhao et al., 2018].

## 2 PRELIMINARIES

In this section, we introduce several important definitions that will be used throughout this paper. Let $c \in \mathbb{R}^d$ and $r \geq 0$; we use $\mathbb{B}(c, r)$ to denote the ball centered at $c$ with radius $r$. Also, given a set $S$ of points in $\mathbb{R}^d$, we use $\text{conv}(S)$ to denote the convex hull of $S$. We use the function $\text{dist}(p, U)$

to measure the shortest distance from a point $p$ to a set $U$, *i.e.*, $\text{dist}(q, U) := \min_{q \in U} ||p - q||$.

**Definition 1** ($k$-center clustering). *Given a set $P$ of $n$ points in $\mathbb{R}^d$ and $k \in \mathbb{Z}^+$, the goal of $k$-center clustering is to find $k$ balls $\mathbb{B}(c_1, r), \cdots, \mathbb{B}(c_k, r)$ with the smallest radius $r$ to cover the set $P$, that is, $P$ is partitioned into $k$ clusters with each cluster being covered by an individual ball, and the radius $r$ is minimized.*

**Remark 1.** *The $k$-center clustering problem can be also defined for any abstract metric, where the only difference is that the Euclidean distance is replaced by the distance defined in the metric. In fact, our proposed sublinear algorithms for $k$-center clustering in this paper can be applied to any abstract metric with the same quality guarantees.*

Let $r_{\text{opt}}$ be the radius of the optimal solution for the $k$-center clustering on $P$. For any solution having a radius $r \leq \lambda r_{\text{opt}}$ with some $\lambda \geq 1$, we call it a "$\lambda$**-approximation**".

**Definition 2** (**convex hull approximation**). *Given a set $P$ of $n$ points in $\mathbb{R}^d$ and an integer $k \geq 1$, the goal of convex hull approximation is to find a subset $P_c \subset P$ with $|P_c| = k$, such that the error, i.e., $\max_{p \in P} \text{dist}(p, \text{conv}(P_c))$ is minimized (so if all the points of $P$ are covered by $\text{conv}(P_c)$, the error is 0).*

**Remark 2.** *In general, we can remove the requirement "$P_c \subset P$", i.e., $P_c$ can contain any points in the space. But we often want $P_c$ to be meaningful or interpretable in practice, and thus it is natural to require it to be a subset of the original input data.*

Similar with $k$-center clustering, we can also define the approximation solution for convex hull approximation. But since the convex hull approximation is much more challenging, we often obtain bi-criteria approximations. Suppose $\alpha, \beta \geq 1$. If letting $\delta_{\text{opt}}$ be the optimal error, a bi-criteria $(\alpha, \beta)$-approximation means that the obtained convex hull has the error $\delta \leq \alpha\delta_{\text{opt}}$ and the number of vertices $k' \leq \beta k$.

**The rest of this paper is organized as follows.** In Section 3, we propose our sublinear time algorithm for $k$-center clustering. In particular, we also consider the practical case that the number of clusters $k$ is not given (Section 3.2). Further, in Section 4 we consider developing sublinear time algorithm for convex hull approximation by extending the idea from Section 3.2. Finally, we present our experiments in Section 5.

## 3 $k$-CENTER CLUSTERING

In this section, we focus on the $k$-center clustering problem. For the sake of completeness, we briefly introduce the aforementioned 2-approximate Gonzalez's algorithm [Gonzalez, 1985] first.

---

[1]The asymptotic notation $\tilde{O}(f) = O(f \cdot \text{polylog}(\frac{kd}{\eta\epsilon}))$, where $\eta \in (0, 1)$ is the parameter controlling the success probability of sampling.

**Gonzalez's algorithm.** It selects an arbitrary point, say $c_1$, from the input $P$ and lets $C = \{c_1\}$. In each of the following $k-1$ iterations, it selects a new point that has the largest distance to $C$ among the points of $P$ and adds it to $C$. Suppose $C = \{c_1, \cdots, c_k\}$, and then $P$ is covered by the $k$ balls $\mathbb{B}(c_1, r), \cdots, \mathbb{B}(c_k, r)$ with $r \leq \min\{\|c_i - c_j\| \mid 1 \leq i \neq j \leq k\}$. It is not difficult to prove that the obtained radius $r \leq 2r_{\text{opt}}$. It is also easy to know that the running time of the Gonzalez's algorithm is $O(knd)$. As mentioned before, a major drawback of the algorithm is the high complexity, especially when $n$ and $d$ are large.

## 3.1 OUR SUBLINEAR ALGORITHM

Our proposed algorithm can be viewed as a randomized version of the Gonzalez's algorithm. The key change is that we randomly select the next point for $C$ in each round, instead of always picking the furthest one. Below we prove that this strategy can achieve the same 2-approximation except for a small error on the number of covering points.

---

**Algorithm 1** SUBLINEAR $k$-CENTER CLUSTERING

**Input:** A set $P$ of $n$ points in $\mathbb{R}^d$, $k \in \mathbb{Z}^+$, and two parameters $\eta, \epsilon \in (0, 1)$.
1. Initially, let $C = \{c_1\}$, where $c_1$ is an arbitrary point picked from $P$; $i = 1$.
2. Repeat the following steps $k - 1$ times:
   (a) Sample a set $Q$ of $\frac{k}{\epsilon} \log \frac{k}{\eta}$ points from $P$ uniformly at random.
   (b) Select the furthest point, say $q_0$, from $Q$ to $C$, i.e., $q_0 = \arg_{q \in Q} \max \mathtt{dist}(q, C)$.
   (c) Let $c_{i+1} = q_0$, $C = C \cup \{c_{i+1}\}$, and $i = i + 1$.
3. Return $C$.

---

**Theorem 1.** *Let $C = \{c_1, \cdots, c_k\}$ be the output from Algorithm 1. With probability at least $1 - \eta$, there exists a subset $\tilde{P} \subset P$ with size $|\tilde{P}| \geq (1 - \epsilon)n$, such that $\tilde{P}$ is covered by $\cup_{i=1}^{k} \mathbb{B}(c_i, 2r_{\text{opt}})$.*

To prove Theorem 1, we need the following claim first.

**Claim 1.** *Let $U$ be a set of elements and $V \subseteq U$ with $\frac{|V|}{|U|} = \tau > 0$. Given $\eta \in (0, 1)$, we uniformly select a set $S$ of elements from $U$ at random. Then if $|S| \geq \frac{1}{\tau} \log \frac{1}{\eta}$, with probability at least $1 - \eta$, $S$ contains at least one element from $V$.*

Actually, the above claim is a folklore result that has been presented in several papers before (such as Ding and Xu [2014]). Since each sampled element falls in $V$ with probability $\tau$, we know that the sample $S$ contains at least one element from $V$ with probability $1 - (1 - \tau)^{|S|}$. If we want to guarantee $1 - (1 - \tau)^{|S|} \geq 1 - \eta$, $|S|$ should be at least $\frac{\log 1/\eta}{\log 1/(1-\tau)} \leq \frac{1}{\tau} \log \frac{1}{\eta}$.

*Proof.* **(of Theorem 1])** To help our analysis, we define $C_1 := \emptyset$, and $C_i := \{c_1, c_2, \cdots, c_{i-1}\}$ for each $i = 2, \cdots, k$ of Algorithm 1. Further, we define

$$P_i := \{p \in P \mid \mathtt{dist}(p, C_i) > \mathtt{dist}(c_i, C_i)\} \qquad (1)$$

for $2 \leq i \leq k$. We also define $F_i$ to be the farthest $\frac{\epsilon}{k}|P|$ points from $P$ to $C_i$. Claim 1 implies that the sample $Q$ should contain at least one point from $F_i$ with probability at least $1 - \frac{\eta}{k}$. If this is true, the selected $c_i$ should come from $F_i$ and thus $|P_i| \leq |F_i|$. Therefore, we have $|P_i| \leq \frac{\epsilon}{k}|P|$ with probability at least $1 - \frac{\eta}{k}$. Through taking the union bound over all the $P_i$s, we have: with probability at least $(1 - \frac{\eta}{k})^{k-1} > 1 - \eta$,

$$\forall i = 2, 3, \cdots, k, \qquad |P_i| < \frac{\epsilon}{k}|P|. \qquad (2)$$

Let $\tilde{P} := P \setminus \cup_{i=2}^{k} P_i$. It is easy to know the size

$$|\tilde{P}| \geq (1 - \frac{\epsilon}{k} \times (k-1))|P| > (1 - \epsilon)n. \qquad (3)$$

Next, we only need to prove that $\tilde{P}$ is covered by $\cup_{i=1}^{k} \mathbb{B}(c_i, 2r_{\text{opt}})$. We present the following lemma first.

**Lemma 1.** *For any point $p \in \tilde{P}$, $\mathtt{dist}(p, C) \leq \min_{1 \leq i \neq i' \leq k} \|c_i - c_{i'}\|$.*

Let $O_1, O_2, \cdots, O_k$ be the $k$ clusters obtained from the optimal solution, *i.e.*, $P = \cup_{i=1}^{k} O_i$ and each cluster $O_i$ can be covered by a ball with radius $r_{\text{opt}}$. We consider two cases. Case **(i)**: $\{c_1, \cdots, c_k\}$ fall into the $k$ clusters $O_1, O_2, \cdots, O_k$ separately. Without loss of generality, we assume $c_i \in O_i$ for $i = 1, 2, \cdots, k$. By using the triangle inequality, we know the input set $P$ is covered by $\cup_{i=1}^{k} \mathbb{B}(c_i, 2r_{\text{opt}})$. Consequently, $\tilde{P}$ is also covered by $\cup_{i=1}^{k} \mathbb{B}(c_i, 2r_{\text{opt}})$.

Case **(ii)**: there exist two points, say $c_{i_a}$ and $c_{i_b}$, of $C$ that belong to one optimal cluster, say $O_l$. Thus $\|c_{i_a} - c_{i_b}\| \leq 2r_{\text{opt}}$. From Lemma 1, we know

$$\forall p \in \tilde{P}, \quad \mathtt{dist}(p, C) \leq \min_{1 \leq i \neq i' \leq k} \|c_i - c_{i'}\|$$
$$\leq \|c_{i_a} - c_{i_b}\| \leq 2r_{\text{opt}}. \quad (4)$$

Hence $\tilde{P}$ is covered by $\cup_{i=1}^{k} \mathbb{B}(c_i, 2r_{\text{opt}})$. $\qquad \square$

*Proof.* **(of Lemma 1])** Suppose Lemma 1 is not true. Then there exist some $p_0 \in \tilde{P}$ and two points $c_{i_1}$ and $c_{i_2} \in C$, such that

$$\mathtt{dist}(p_0, C) > \|c_{i_1} - c_{i_2}\|. \qquad (5)$$

Without loss of generality, we assume $i_1 < i_2$. Since $\|c_{i_1} - c_{i_2}\| \geq \mathtt{dist}(c_{i_2}, C_{i_2})$, the inequality (5) implies

$$\mathtt{dist}(p_0, C) > \mathtt{dist}(c_{i_2}, C_{i_2}). \qquad (6)$$

So from (1) we know $p_0 \in P_{i_2}$, which is in contradiction with the assumption $p_0 \in \tilde{P} = P \setminus \cup_{i=2}^{k} P_i$. $\qquad \square$

**Time complexity.** It is easy to see that the time complexity of Algorithm 1 is independent of $n$. It takes $k$ rounds, and each round needs to compute the distances from the sampled $\frac{k}{\epsilon} \log \frac{k}{\eta}$ points to $C$. So the total complexity is $O(k \times \frac{k}{\epsilon} \log \frac{k}{\eta} \times kd) = O(\frac{k^3}{\epsilon} d \log \frac{k}{\eta})$.

In some scenarios, we may not be able to access the whole data, *e.g.,* due to privacy preserving or the challenge of data acquisition. Instead, we may be only allowed to take a small sample each time. Specifically, we assume the data is a (continuous or discrete) probability distribution with the probability density function $f$ in $\Omega \subset \mathbb{R}^d$, where $\int_{p \in \Omega} f(p)\mathrm{d}p = 1$; the function $f$ can be hid and we only assume that there is an oracle to sample data based on $f$. Obviously, it is prohibitive to directly run the Gonzalez's algorithm in such scenario. On the other hand, our proposed Algorithm 1 can be naturally applied to solve this problem because it only takes a random sample in each round. The following result is a straightforward extension of Theorem 1.

**Corollary 1.** *We run Algorithm 1 on a (continuous or discrete) probability distribution over $\Omega$; each sampled point is taken by an oracle based on the probability density function $f$. With probability at least $1 - \eta$, there exists a subset $\tilde{\Omega} \subset \Omega$ with the integral $\int_{p \in \tilde{\Omega}} f(p)\mathrm{d}p \geq 1 - \epsilon$, such that $\tilde{\Omega}$ is covered by $\cup_{i=1}^{k} \mathbb{B}(c_i, 2r_{\mathrm{opt}})$.*

## 3.2 WHEN $k$ IS NOT GIVEN

In many real scenarios, the number of clusters $k$ is often not given. For instance, we may only have a threshold $r_0 > 0$ for the radius; so we just try to perform the $k$-center clustering algorithm for different values of $k$ until the obtained radius is no larger than $r_0$. The reader may realize that this problem is related to the well known *geometric set cover* problem [Brönnimann and Goodrich, 1995, Agarwal and Pan, 2020]; however, existing geometric set cover algorithms often have large (super linear) running time and can only handle low dimensional case. Actually, the geometric set cover problem is NP-hard and has only constant factor approximation in 2D plane (the problem is even harder in high dimensions).

In this paper, we simplify the problem and consider a practical approach: using the Gonzalez's algorithm to achieve our goal. Suppose the given set $P$ can be covered by $\tilde{k} \in \mathbb{Z}^+$ balls with radius $r_0/2$ (*i.e.,* $\tilde{k}$ is the value that the optimal radius of $\tilde{k}$-center clustering on $P$ is no larger than $r_0/2$). Then, if we just run the Gonzalez's algorithm iteratively, the resulting radius will reach $r_0$ within at most $\tilde{k}$ rounds (because it is a 2-approximation algorithm). **Now we discuss how to implement this procedure in sublinear time.** We cannot directly adapt this procedure to our sublinear Algorithm 1, due to the following two issues. **(1)** The sample size $\frac{k}{\epsilon} \log \frac{k}{\eta}$ in step 2(a) depends on a given $k$ (also note that Algorithm 1 is a randomized algorithm and its success

probability depends on the sample size); **(2)** we do not know when to terminate if $k$ is not given.

To resolve these two issues, we introduce a **stratified sampling method**. Let $k_0 \geq 1$ be any fixed constant. Imagine we run step 2(a)-2(c) of Algorithm 1 iteratively. We partition the process into different phases and modify the sample size in step 2(a) for each phase accordingly:

- **Phase** $t = 0$**:** for $i = 1, 2, \cdots, k_0$, we set $|Q| = \frac{2k_0}{\epsilon} \log \frac{k_0}{\eta}$.

- **Phase** $t \geq 1$**:** for $i = \sum_{s=0}^{t-1} 2^s k_0 + 1, \sum_{s=0}^{t-1} 2^s k_0 + 2, \cdots, \sum_{s=0}^{t} 2^s k_0$, we set $|Q| = 2^{2t} \frac{2k_0}{\epsilon} \log \frac{2^t k_0}{\eta}$.

So phase $t$ contains $2^t k_0$ iterations. The sample size also increases from phase $t$ to phase $t + 1$.

For completeness, we also need to set the stopping condition. Suppose $r_0 > 0$ is the given threshold. At the end of each $i$-th iteration, we take a sample $S$ from $P$ uniformly at random, and compute the ratio

$$\tau = \frac{\left| S \setminus \left( \cup_{l=1}^{i} \mathbb{B}(c_l, r_0) \right) \right|}{|S|}. \tag{7}$$

The following lemma introduces an **oracle** that can help us to decide when to terminate.

**Lemma 2.** *Suppose $\eta_0 \in (0, 1)$. We set the sample size $|S| \geq \frac{12}{\eta_0 \epsilon} \log \frac{2}{\eta_0}$. With probability at least $1 - \eta_0$, the following oracle returns the correct answer: if $\tau \leq \frac{3}{2}\epsilon$, return "$\left| P \setminus \left( \cup_{l=1}^{i} \mathbb{B}(c_l, r_0) \right) \right| \leq 3\epsilon n$"; else, return "$\left| P \setminus \left( \cup_{l=1}^{i} \mathbb{B}(c_l, r_0) \right) \right| > \epsilon n$".*

*Proof.* For convenience, we use $\tilde{\epsilon}$ to denote the ratio $\frac{\left| P \setminus \left( \cup_{l=1}^{i} \mathbb{B}(c_l, r_0) \right) \right|}{n}$. We consider two cases: (i) $\tilde{\epsilon} \leq \eta_0 \epsilon$ and (ii) $\tilde{\epsilon} > \eta_0 \epsilon$. For case (i), $\left| P \setminus \left( \cup_{l=1}^{i} \mathbb{B}(c_l, r_0) \right) \right| = \tilde{\epsilon} n \leq \eta_0 \epsilon n < 3\epsilon n$. Due to the Markov's inequality, we know that $\tau \leq \frac{1}{\eta_0} \times \eta_0 \epsilon = \epsilon < \frac{3}{2}\epsilon$ with probability at least $1 - \eta_0$. Thus, it returns "$\left| P \setminus \left( \cup_{l=1}^{i} \mathbb{B}(c_l, r_0) \right) \right| \leq 3\epsilon n$" which is a correct answer, with probability at least $1 - \eta_0$. So we focus on the second case below.

We use the Chernoff bound [Alon and Spencer, 2004]. Define $|S|$ random variables $\{y_1, \cdots, y_{|S|}\}$: for each $1 \leq j \leq |S|$, $y_j = 1$ if the $j$-th sampled element falls in $P \setminus \left( \cup_{l=1}^{i} \mathbb{B}(c_l, r_0) \right)$, otherwise, $y_j = 0$. So $E[y_j] = \tilde{\epsilon}$ for each $y_j$. As a consequence, we have

$$\mathbf{Pr}\left( \left| \sum_{j=1}^{|S|} y_j - \tilde{\epsilon}|S| \right| \leq \frac{1}{2}\tilde{\epsilon}|S| \right) \geq 1 - 2e^{-\frac{\tilde{\epsilon}}{12}|S|}. \tag{8}$$

Since we assume $\tilde{\epsilon} > \eta_0 \epsilon$, if $|S| \geq \frac{12}{\eta_0 \epsilon} \log \frac{2}{\eta_0}$, the above (8) implies that with probability at least $1 - \eta_0$, $\left| \sum_{j=1}^{|S|} y_j - \right.$

$\tilde\epsilon|S|| \le \frac{1}{2}\tilde\epsilon|S|$, *i.e.*,

$$\tau = \frac{\sum_{j=1}^{|S|} y_j}{|S|} \in [\frac{1}{2}\tilde\epsilon, \frac{3}{2}\tilde\epsilon]. \tag{9}$$

Therefore, if $\tau \le \frac{3}{2}\epsilon$, we know $\frac{1}{2}\tilde\epsilon \le \frac{3}{2}\epsilon$ from (9), and it implies $\tilde\epsilon \le 3\epsilon$. Otherwise, we know $\frac{3}{2}\tilde\epsilon > \frac{3}{2}\epsilon$ and it implies $\tilde\epsilon > \epsilon$. $\qquad\square$

Now, we are ready to present our algorithm for the case without knowing $\tilde k$. Let $i_{\texttt{ter}}$ be the size of $C$ when Algorithm 2 terminates. To evaluate the performance of the algorithm, we need to compare $i_{\texttt{ter}}$ with $\tilde k$ and investigate the number of points that are covered by $\cup_{j=1}^{i_{\texttt{ter}}}\mathbb{B}(c_j, r_0)$.

---

**Algorithm 2** SUBLINEAR $k$-CENTER CLUSTERING II

---

**Input:** A set $P$ of $n$ points in $\mathbb{R}^d$, a threshold $r_0 > 0$, an arbitrary constant integer $k_0 \in \mathbb{Z}^+$, and two parameters $\eta, \epsilon \in (0, 1)$.

1. Initially, let $C = \{c_1\}$, where $c_1$ is an arbitrary point picked from $P$; $t = i = 0$.

2. Repeat the following steps as the stratified sampling procedure:

   (a) Take a sample $S$ from $P$ uniformly at random, where $|S| = \frac{12}{\eta_0\epsilon}\log\frac{2}{\eta_0}$ and $\eta_0 = \frac{\eta}{2^{2t}k_0}$.

   (b) Repeat the following steps $2^t k_0$ times (*i.e.*, phase $t$):

      i. Randomly pick a set $Q$ from $P$, where $|Q| = 2^{2t}\frac{2k_0}{\epsilon}\log\frac{2^{2t}k_0}{\eta}$.

      ii. Let $q_0$ be the furthest point from $Q$ to $C$, *i.e.*, $q_0 = \arg_{q\in Q}\max\texttt{dist}(q, C)$.

      iii. Let $c_{i+1} = q_0$, $C = C\cup\{c_{i+1}\}$, and $i = i+1$.

      iv. Apply Lemma 2 as the oracle (using the sample $S$ from step 2(a)) to determine whether to terminate: if it returns "$\left|P \setminus \left(\cup_{l=1}^{i}\mathbb{B}(c_l, r_0)\right)\right| \le 3\epsilon n$", stop the algorithm, set $i_{\texttt{ter}} = i$, and return $C$.

   (c) $t = t+1$.

---

**Theorem 2.** *Let $C = \{c_1, \cdots, c_{i_{\texttt{ter}}}\}$ be the output from Algorithm 2. With probability at least $1 - 4\eta$, $i_{\texttt{ter}} \le \tilde k$, and there exists a subset $\tilde P \subset P$ with size $|\tilde P| \ge (1-3\epsilon)n$, such that $\tilde P$ is covered by $\cup_{j=1}^{i_{\texttt{ter}}}\mathbb{B}(c_j, r_0)$.*

*Proof.* To prove Theorem 2, we first imagine the "fancied" scenario that $\tilde k$ is given: we just run Algorithm 1 with $k = \tilde k$ and $|Q| = \frac{\tilde k}{\epsilon}\log\frac{\tilde k}{\eta}$. Recall the proof of Theorem 1, where we define a sequence of subsets $P_2, P_3, \cdots, P_{\tilde k}$ and define $\tilde P = P \setminus \cup_{i=2}^{\tilde k}P_i$. To guarantee $|\tilde P| \ge (1-\epsilon)n$, we prove

that each $P_i$ contains at most $\frac{\epsilon}{\tilde k}n$ points. For Algorithm 2, we also define a sequence of subsets $P_2, P_3, \cdots, P_{\tilde k}$ (by using (1)), but we need to modify their sizes. At each $t$-th phase, since we have the sample size $2^{2t}\frac{2k_0}{\epsilon}\log\frac{2^{2t}k_0}{\eta}$, by using the similar idea from the proof of Theorem 1 we know that the size

$$|P_i| \le \frac{\epsilon}{2^{2t} \times 2k_0}n, \text{ with probability } \ge 1 - \frac{\eta}{2^{2t}k_0}. \tag{10}$$

Suppose we run Algorithm 2 until $i = \tilde k$. Let $t_0$ be the total number of phases that the algorithm takes. Consequently, we have

$$\frac{\left|\cup_{i=2}^{\tilde k}P_i\right|}{n} \le \frac{\epsilon}{2k_0}\times k_0 + \frac{\epsilon}{2^2\times 2k_0}\times 2k_0$$
$$+\cdots + \frac{\epsilon}{2^{2t_0}\times 2k_0}\times 2^{t_0}k_0$$
$$= \frac{\epsilon}{2}(1 + \frac{1}{2} + \cdots + \frac{1}{2^{t_0}}) \le \epsilon. \tag{11}$$

So we can still guarantee $|\tilde P| = |P \setminus \cup_{i=2}^{\tilde k}P_i| \ge (1-\epsilon)n$. Furthermore, the total success probability is at least

$$(1 - \frac{\eta}{k_0})^{k_0} \times (1 - \frac{\eta}{2^2 k_0})^{2k_0}$$
$$\times \cdots \times (1 - \frac{\eta}{2^{2(t_0-1)}k_0})^{2^{t_0-1}k_0}$$
$$> (1-\eta) \times (1 - \frac{\eta}{2}) \times \cdots \times (1 - \frac{\eta}{2^{t_0-1}})$$
$$> 1 - (1 + \frac{1}{2} + \cdots + \frac{1}{2^{t_0-1}})\eta > 1 - 2\eta. \tag{12}$$

The remaining issue is that we do not know the value of $\tilde k$ in reality (in other words, we do not know when to terminate the algorithm). Therefore, we apply Lemma 2 as an oracle in step 2(b)(iv), where the success probability for each time is $1-\eta_0 = 1 - \frac{\eta}{2^{2t}k_0}$. When it returns "$\left|P\setminus\left(\cup_{l=1}^{i}\mathbb{B}(c_l, r_0)\right)\right| > \epsilon n$", we know that the algorithm needs to continue. We stop the algorithm when it returns "$\left|P \setminus \left(\cup_{l=1}^{i}\mathbb{B}(c_l, r_0)\right)\right| \le 3\epsilon n$". Since we relax the error of covering number to be $3\epsilon > \epsilon$, we know that $i_{\texttt{ter}}$ should be no larger than $\tilde k$. By using the similar idea of (12), we can obtain the overall success probability of the oracle that is at least

$$1 - 2\eta. \tag{13}$$

Combining (12) and (13), the overall success probability of Algorithm 2 is at least $1 - 4\eta$. $\qquad\square$

**The time complexity of Algorithm 2.** We analyze the runtime for each phase. We set $k_0 \ge 2$ to be a constant integer. At the $t$-th phase, step (b)(i)-(iii) take $O(\frac{2^{3t}}{\epsilon}\log\frac{2^{2t}}{\eta}d)$ time; step (b)(iv) takes $O(\frac{2^{2t}}{\eta\epsilon}\log\frac{2^{2t}}{\eta}d)$ time. Also, the phase repeats step (b)(i)-(iv) $O(2^t)$ times. Thus, the $t$-th phase takes $O((2^t + \frac{1}{\eta})\frac{2^{3t}}{\epsilon}\log\frac{2^{2t}}{\eta}d)$ time. Let $t_0$ be the total number of phases. Then we can calculate the bounds for $\tilde k$:

$$\sum_{s=0}^{t_0-1}2^s k_0 < \tilde k \le \sum_{s=0}^{t_0}2^s k_0, \tag{14}$$

which implies $t_0 \leq \log \frac{\tilde{k}}{k_0} + 1 \leq \log \tilde{k}$. So the total time complexity of Algorithm 2 is $O((\tilde{k} + \frac{1}{\eta})\frac{\tilde{k}^3}{\epsilon}\log\frac{\tilde{k}}{\eta}d)$. Compared with the case that $\tilde{k}$ is given, the runtime is increased by only a factor $(\tilde{k} + \frac{1}{\eta})$ (the runtime of Algorithm 1 is $O(\frac{\tilde{k}^3}{\epsilon}\log\frac{\tilde{k}}{\eta}d)$).

We also have the following result for Algorithm 2 which is similar with Corollary 1.

**Corollary 2.** *We run Algorithm 2 on a (continuous or discrete) probability distribution over $\Omega$; each sampled point is taken by an oracle based on the probability density function $f$. With probability at least $1 - 4\eta$, $i_{\mathtt{ter}} \leq \tilde{k}$, and there exists a subset $\tilde{\Omega} \subset \Omega$ with the integral $\int_{p \in \tilde{\Omega}} f(p)\mathrm{d}p \geq 1 - 3\epsilon$, such that $\tilde{\Omega}$ is covered by $\cup_{i=1}^{i_{\mathtt{ter}}} \mathbb{B}(c_i, r_0)$.*

# 4 CONVEX HULL APPROXIMATION IN HIGH DIMENSIONS

Blum et al. [2019] introduced a simple greedy convex hull approximation algorithm that is similar in spirit to the Gonzalez's algorithm for $k$-center clustering. Given an instance $P \subset \mathbb{R}^d$, it also maintains a set $C$ that contains an arbitrarily selected $p \in P$ at the beginning. In each round, the algorithm always selects the farthest point to $\mathtt{conv}(C)$ and adds it to $C$, until some specified stopping condition is satisfied. For ease of presentation, we assume that $P$ is contained in a unit ball of $\mathbb{R}^d$. The algorithm yields a bi-criteria approximate result: given an error parameter $\delta \in (0, 1)$, suppose $k_{\mathtt{opt}} = \min\{k \mid Q \subset P, |Q| = k, \max_{p \in P} \mathtt{dist}(p, \mathtt{conv}(Q)) \leq \delta\}$; the algorithm can yield a subset $C \subset P$ such that

$$|C| = O(k_{\mathtt{opt}}/\delta^{2/3})$$
$$\mathtt{dist}(p, \mathtt{conv}(C)) \leq 8\delta^{1/3} + \delta, \forall p \in P. \quad (15)$$

We consider applying our previous sampling idea to implement this convex hull approximation algorithm in sublinear time. Here, we have the same issue as Section 3.2, that is, we do not know the exact value of $k_{\mathtt{opt}}$ so that we cannot determine the sample size in each iteration and when to terminate. Thus we apply the same stratified sampling method. We also use Lemma 2 as the oracle to determine whether the stopping condition is satisfied.

A minor technical issue for implementation is that it is costly to compute the distance from a given point to a convex hull (it needs to solve a quadratic programming for achieving the exact result); instead we can apply the Gilbert's algorithm [Gilbert, 1966, Gärtner and Jaggi, 2009] or some other variants like the Triangle algorithm [Awasthi et al., 2020] to compute an approximate solution efficiently. Thus, we need another small parameter $\xi \in (0, 1)$ to indicate the approximation error induced by this step. Compared with the ratio

"$\tau$" for $k$-center clustering, we add an extra factor $(1 + \xi)$ to $\tau$ below.

Let $r_0 = 8\delta^{1/3} + \delta$. We use $C_i$ to denote the set of selected vertices $\{c_1, \cdots, c_i\}$ at the first $i$ rounds. For convenience, we use $\mathtt{conv}(U, r)$ to denote the set

$$\{p \mid p \in \mathbb{R}^d, \mathtt{dist}(p, \mathtt{conv}(U)) \leq r\}$$

for any given set $U$ and $r \geq 0$. Then, we compute the ratio

$$\tau = \frac{\left| S \setminus \mathtt{conv}(C_i, (1 + \xi)r_0) \right|}{|S|}. \quad (16)$$

Similar to the case of $k$-center clustering, the following lemma introduces an **oracle** that can help us to decide when to terminate (the proof is almost identical to that of Lemma 2).

---

**Algorithm 3** SUBLINEAR CONVEX HULL APPROXIMATION

**Input:** A set $P$ of $n$ points in $\mathbb{R}^d$, a threshold $r_0 > 0$, an arbitrary constant integer $k_0 \in \mathbb{Z}^+$, and three parameters $\eta, \epsilon, \xi \in (0, 1)$.

1. Initially, let $C = \{c_1\}$, where $c_1$ is an arbitrary point picked from $P$; $t = i = 0$.

2. Repeat the following steps as the stratified sampling procedure:

   (a) Take a sample $S$ from $P$ uniformly at random, where $|S| = \frac{12}{\eta_0 \epsilon} \log \frac{2}{\eta_0}$ and $\eta_0 = \frac{\eta}{2^{2t} k_0}$.

   (b) Repeat the following steps $2^t k_0$ times (*i.e.*, phase $t$):

      i. Randomly pick a set $Q$ from $P$, where $|Q| = 2^{2t} \frac{2k_0}{\epsilon} \log \frac{2^{2t} k_0}{\eta}$.

      ii. Select the $(1 + \xi)$-approximate furthest point, say $q_0$, from $Q$ to $\mathtt{conv}(C)$ via the algorithm of Gärtner and Jaggi [2009].

      iii. Let $c_{i+1} = q_0$, $C = C \cup \{c_{i+1}\}$, and $i = i+1$.

      iv. Apply Lemma 3 as the oracle (using the sample $S$ from step 2(a)) to determine whether to terminate: if it returns "$\left| P \setminus \mathtt{conv}(C_i, (1 + \xi)r_0) \right| \leq 3\epsilon n$", stop the algorithm, set $i_{\mathtt{ter}} = i$, and return $C$.

   (c) $t = t + 1$.

---

**Lemma 3.** *Suppose $\eta_0 \in (0, 1)$. We set the sample size $|S| \geq \frac{12}{\eta_0 \epsilon} \log \frac{2}{\eta_0}$. With probability at least $1 - \eta_0$, the following oracle returns the correct answer: if $\tau \leq \frac{3}{2}\epsilon$, return "$\left| P \setminus \mathtt{conv}(C_i, (1 + \xi)r_0) \right| \leq 3\epsilon n$"; else, return "$\left| P \setminus \mathtt{conv}(C_i, (1 + \xi)r_0) \right| > \epsilon n$".*

**Theorem 3.** *Let $C = \{c_1, \cdots, c_{i_{\texttt{ter}}}\}$ be the output from Algorithm 3. Let $\tilde{k}$ be the number of vertices returned by the greedy selection algorithm [Blum et al., 2019] (see (15)). With probability at least $1 - 4\eta$, $i_{\texttt{ter}} \leq \tilde{k}$, and there exists a subset $\tilde{P} \subset P$ with size $|\tilde{P}| \geq (1 - 3\epsilon)n$, such that $\tilde{P}$ is covered by $\texttt{conv}(C, (1 + \xi)r_0)$.*

**Time complexity.** The computation for the time complexity is similar with that for $k$-center clustering in Section 3.2, where the only difference is that we have to compute the $(1 + \xi)$-approximate polytope distance from each sampled point to $C$ in each iteration. From the analysis of Gärtner and Jaggi [2009], we know it takes $O(\frac{1}{\xi\delta^2}|C|d)$ time. The total complexity of our convex hull algorithm is $O((\tilde{k} + \frac{1}{\eta})\frac{\tilde{k}^3}{\epsilon\xi\delta^2}\log\frac{\tilde{k}}{\eta}d)$.

## 5 EXPERIMENTAL RESULTS

All the experiments were conducted on an Ubuntu workstation with 2.40GHz Intel(R) Xeon(R) CPU E5-2680 and 256GB main memory. The algorithms were implemented in MATLAB R2019b. For each instance, we repeat the experiment 10 times and report the average results with their standard deviations.

We consider several baseline methods including the 2-approximate GONZALEZ [Gonzalez, 1985] and the recently proposed streaming $k$-center clustering algorithm CPP [Ceccarello et al., 2019]; we also compare our algorithms with the uniform sampling method Huang et al. [2018] that is denoted as UNIFORM-$r$, where $r$ denotes the sampling rate (*e.g.,* UNIFORM-0.1 means we take $10\%$ points from the input uniformly at random).

We run our proposed Algorithm 1 and Algorithm 2 on the real image dataset **CIFAR-10** [Krizhevsky, 2009] which consists of $60,000$ color images with each image being represented by a 3072-dimensional vector. In Figure 1, we can see that our Algorithm 1 runs significantly faster than the other methods. Also it is worth emphasizing that in our evaluation, we compute the radius for covering all the input points, rather than excluding the farthest $\epsilon n$ points as the theoretical analysis in Theorem 1. We can see that our algorithm and GONZALEZ can achieve very close radii, even though we did not exclude the farthest $\epsilon n$ points.

In Figure 2, we illustrate the results of Algorithm 2. Similar with Figure 1, we can see that our algorithm runs much faster than GONZALEZ. An interesting observation is that our algorithm returns much less centers than GONZALEZ for a fixed radius threshold $r_0$ (Figure 2 (c)). We believe one possible reason is that our random sampling approach is more likely to select a point closer to the optimal ball center, and thus the obtained radius can decrease faster, while the greedy selection of GONZALEZ always selects the most "extreme" point which could be far to the optimal ball center.

We also run the algorithms on another real dataset **MNIST** [LeCun et al., 1998]; it contains $n = 60,000$ handwritten digit images from 0 to 9, where each image is represented by a 784-dimensional vector. To illustrate the scalability of our algorithms for large-scale data, we enlarge **MNIST** by 6 times; namely, for each image vector, we generate 5 copies and add small Gaussian noises to them. The results are shown in Figure 3 and 4.

Due to the space limit, we place the experimental results for convex hull approximation to our full version.

## 6 CONCLUSION

In this paper, we propose the sublinear algorithms for greedy selection methods. Following this work, there are also several interesting problems deserving to study in future. For example, in our experiments we observe that our random sampling based approach can achieve very close radii with the vanilla greedy selection approach GONZALEZ (even without excluding the farthest $\epsilon n$ points). So we expect to have a strict analysis on this phenomenon in theory, *e.g.,* adding some reasonable assumption to the data distribution from the perspective of *beyond worst-case analysis* [Roughgarden, 2019].

## 7 ACKNOWLEDGEMENTS

The authors would like to thank the anonymous reviewers for their helpful discussions and suggestions on improving this paper. This work was supported in part by National Key R & D Program of China No. 2021YFA1000900.

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

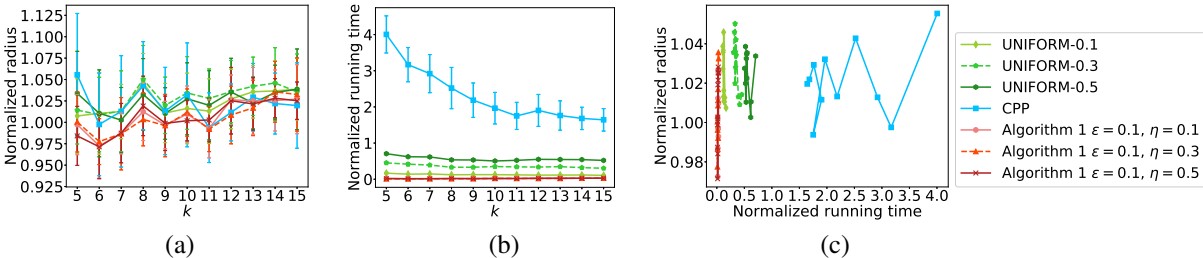

Figure 1: The experimental performances on **CIFAR-10** for the case that $k$ is given. All the results (radius and runtime) are respectively normalized over the results obtained by GONZALEZ. In (c), we show the radius obtained versus runtime for different values of $k$.

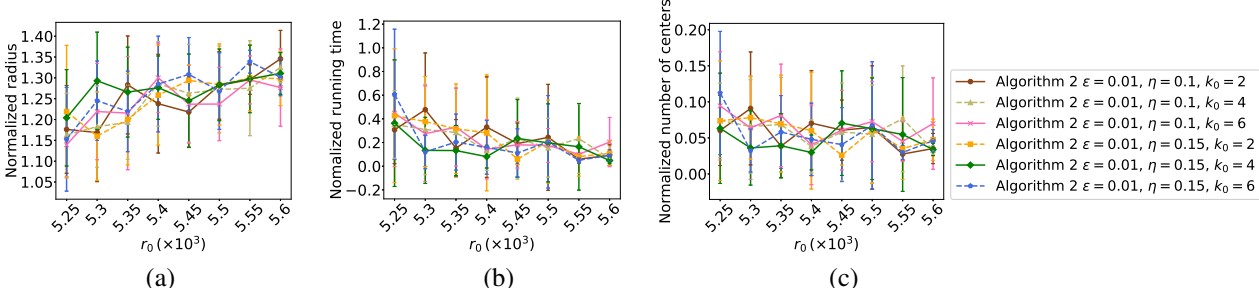

Figure 2: The experimental performances on **CIFAR-10** for the case that a radius threshold $r_0$ is given. All the results (radius, runtime, and the number of returned centers) are respectively normalized over the results obtained by GONZALEZ.

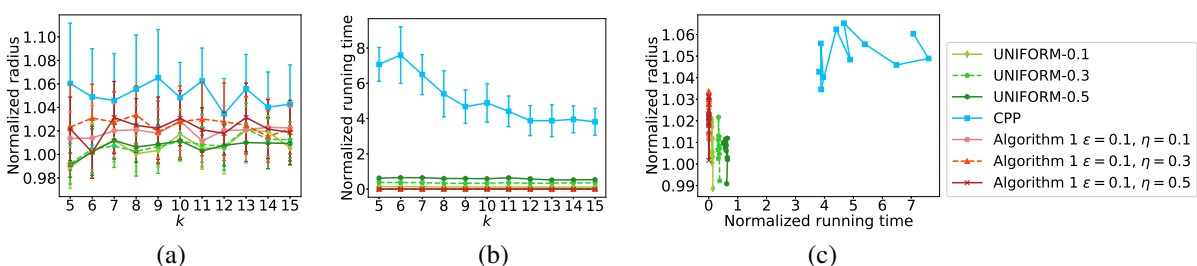

Figure 3: The experimental performances on **MNIST** for the case that $k$ is given. All the results (radius and runtime) are respectively normalized over the results obtained by GONZALEZ. In (c), we show the radius obtained versus runtime for different values of $k$.

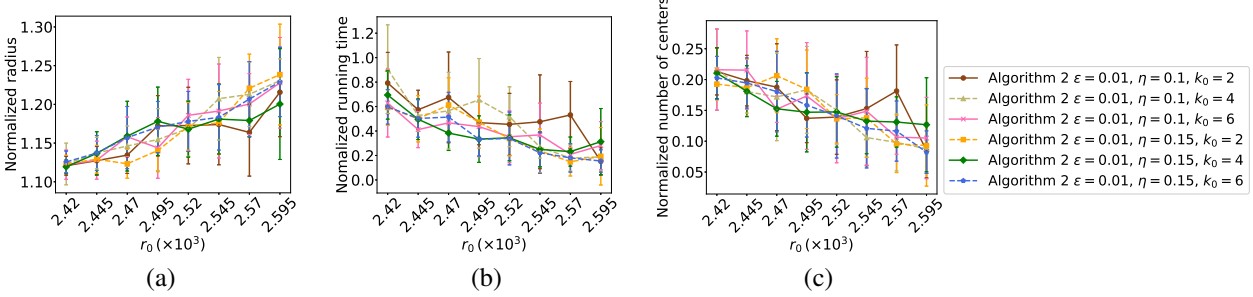

Figure 4: The experimental performances on **MNIST** for the case that a radius threshold $r_0$ is given. All the results (radius, runtime, and the number of returned centers) are respectively normalized over the results obtained by GONZALEZ.

and Michael Zhu. A practical algorithm for topic modeling with provable guarantees. In *Proceedings of the 30th International Conference on Machine Learning, ICML 2013, Atlanta, GA, USA, 16-21 June 2013*, volume 28 of *JMLR Workshop and Conference Proceedings*, pages 280–288. JMLR.org, 2013.

Pranjal Awasthi, Bahman Kalantari, and Yikai Zhang. Robust vertex enumeration for convex hulls in high dimensions. *Ann. Oper. Res.*, 295(1):37–73, 2020.

Mihai Bādoiu, Sariel Har-Peled, and Piotr Indyk. Approximate clustering via core-sets. In *Proceedings of the thiry-fourth annual ACM symposium on Theory of computing*, pages 250–257, 2002.

Avrim Blum, Sariel Har-Peled, and Benjamin Raichel. Sparse approximation via generating point sets. *ACM Transactions on Algorithms*, 15(3):32, 2019.

Hervé Brönnimann and Michael T. Goodrich. Almost optimal set covers in finite vc-dimension. *Discret. Comput. Geom.*, 14(4):463–479, 1995.

Matteo Ceccarello, Andrea Pietracaprina, and Geppino Pucci. Solving k-center clustering (with outliers) in mapreduce and streaming, almost as accurately as sequentially. *PVLDB*, 12(7):766–778, 2019.

Moses Charikar, Liadan O'Callaghan, and Rina Panigrahy. Better streaming algorithms for clustering problems. In *Proceedings of the thirty-fifth annual ACM symposium on Theory of computing*, pages 30–39. ACM, 2003.

Moses Charikar, Chandra Chekuri, Tomás Feder, and Rajeev Motwani. Incremental clustering and dynamic information retrieval. *SIAM J. Comput.*, 33(6):1417–1440, 2004.

Cody Coleman, Christopher Yeh, Stephen Mussmann, Baharan Mirzasoleiman, Peter Bailis, Percy Liang, Jure Leskovec, and Matei Zaharia. Selection via proxy: Efficient data selection for deep learning. In *8th International Conference on Learning Representations, ICLR 2020, Addis Ababa, Ethiopia, April 26-30, 2020*. OpenReview.net, 2020.

Thomas H. Cormen, Charles E. Leiserson, Ronald L. Rivest, and Clifford Stein. *Introduction to Algorithms, Third Edition*. The MIT Press, 3rd edition, 2009. ISBN 0262033844, 9780262033848.

Sanjoy Dasgupta and Anupam Gupta. An elementary proof of a theorem of johnson and lindenstrauss. *Random Structures & Algorithms*, 22(1):60–65, 2003.

M. de Berg, O. Cheong, M. van Kreveld, and O. Schwarzkopf. *Computational Geometry: Algorithms and Applications, (Third Edition)*. Springer-Verlag, USA, 2008. ISBN 978-3-540-77973-5.

Hu Ding and Jinhui Xu. Sub-linear time hybrid approximations for least trimmed squares estimator and related problems. In *Proceedings of the International Symposium on Computational geometry (SoCG)*, page 110, 2014.

David L. Donoho and Victoria Stodden. When does non-negative matrix factorization give a correct decomposition into parts? In Sebastian Thrun, Lawrence K. Saul, and Bernhard Schölkopf, editors, *Advances in Neural Information Processing Systems 16 [Neural Information Processing Systems, NIPS 2003, December 8-13, 2003, Vancouver and Whistler, British Columbia, Canada]*, pages 1141–1148. MIT Press, 2003.

Ahmed K. Farahat, Ahmed Elgohary, Ali Ghodsi, and Mohamed S. Kamel. Greedy column subset selection for large-scale data sets. *Knowl. Inf. Syst.*, 45(1):1–34, 2015.

Tomás Feder and Daniel H. Greene. Optimal algorithms for approximate clustering. In Janos Simon, editor, *Proceedings of the 20th Annual ACM Symposium on Theory of Computing, May 2-4, 1988, Chicago, Illinois, USA*, pages 434–444. ACM, 1988.

Bernd Gärtner and Martin Jaggi. Coresets for polytope distance. In *Proceedings of the twenty-fifth annual symposium on Computational geometry*, pages 33–42, 2009.

Rong Ge and Ankur Moitra. Topic models and nonnegative matrix factorization. In Tim Roughgarden, editor, *Beyond the Worst-Case Analysis of Algorithms*, pages 445–464. Cambridge University Press, 2020.

Elmer G. Gilbert. An iterative procedure for computing the minimum of a quadratic form on a convex set. *SIAM Journal on Control*, 4(1):61–80, 1966.

Teofilo F. Gonzalez. Clustering to minimize the maximum intercluster distance. *Theoretical Computer Science*, 38: 293–306, 1985. ISSN 0304-3975.

Sudipto Guha. Tight results for clustering and summarizing data streams. In Ronald Fagin, editor, *Database Theory - ICDT 2009, 12th International Conference, St. Petersburg, Russia, March 23-25, 2009, Proceedings*, volume 361 of *ACM International Conference Proceeding Series*, pages 268–275. ACM, 2009.

Sariel Har-Peled and Manor Mendel. Fast construction of nets in low-dimensional metrics and their applications. *SIAM Journal on Computing*, 35(5):1148–1184, 2006.

Avinatan Hassidim and Yaron Singer. Robust guarantees of stochastic greedy algorithms. In Doina Precup and Yee Whye Teh, editors, *Proceedings of the 34th International Conference on Machine Learning, ICML 2017, Sydney, NSW, Australia, 6-11 August 2017*, volume 70 of *Proceedings of Machine Learning Research*, pages 1424–1432. PMLR, 2017.

Dorit S Hochbaum and David B Shmoys. A best possible heuristic for the k-center problem. *Mathematics of operations research*, 10(2):180–184, 1985.

Lingxiao Huang, Shaofeng Jiang, Jian Li, and Xuan Wu. Epsilon-coresets for clustering (with outliers) in doubling metrics. In *2018 IEEE 59th Annual Symposium on Foundations of Computer Science (FOCS)*, pages 814–825. IEEE, 2018.

Piotr Indyk, Sepideh Mahabadi, Mohammad Mahdian, and Vahab S. Mirrokni. Composable core-sets for diversity and coverage maximization. In Richard Hull and Martin Grohe, editors, *Proceedings of the 33rd ACM SIGMOD-SIGACT-SIGART Symposium on Principles of Database Systems, PODS'14, Snowbird, UT, USA, June 22-27, 2014*, pages 100–108. ACM, 2014.

Alex Krizhevsky. Learning multiple layers of features from tiny images. Technical report, 2009.

Yann LeCun, Léon Bottou, Yoshua Bengio, and Patrick Haffner. Gradient-based learning applied to document recognition. *Proceedings of the IEEE*, 86(11):2278–2324, 1998.

Richard Matthew McCutchen and Samir Khuller. Streaming algorithms for k-center clustering with outliers and with anonymity. In *Approximation, Randomization and Combinatorial Optimization. Algorithms and Techniques*, pages 165–178. Springer, 2008.

Baharan Mirzasoleiman, Ashwinkumar Badanidiyuru, Amin Karbasi, Jan Vondrák, and Andreas Krause. Lazier than lazy greedy. In Blai Bonet and Sven Koenig, editors, *Proceedings of the Twenty-Ninth AAAI Conference on Artificial Intelligence, January 25-30, 2015, Austin, Texas, USA*, pages 1812–1818. AAAI Press, 2015.

George L Nemhauser, Laurence A Wolsey, and Marshall L Fisher. An analysis of approximations for maximizing submodular set functions—i. *Mathematical programming*, 14(1):265–294, 1978.

Christopher Painter-Wakefield and Ronald Parr. Greedy algorithms for sparse reinforcement learning. In *Proceedings of the 29th International Conference on Machine Learning, ICML 2012, Edinburgh, Scotland, UK, June 26 - July 1, 2012*. icml.cc / Omnipress, 2012.

Tim Roughgarden. Beyond worst-case analysis. *Commun. ACM*, 62(3):88–96, 2019.

Maximilian Schleich, Dan Olteanu, Mahmoud Abo Khamis, Hung Q. Ngo, and XuanLong Nguyen. Learning models over relational data: A brief tutorial. *CoRR*, abs/1911.06577, 2019.

Ozan Sener and Silvio Savarese. Active learning for convolutional neural networks: A core-set approach. In *International Conference on Learning Representations*, 2018.

Joel A. Tropp. Greed is good: algorithmic results for sparse approximation. *IEEE Trans. Inf. Theory*, 50(10):2231–2242, 2004.

Zhuoyue Zhao, Robert Christensen, Feifei Li, Xiao Hu, and Ke Yi. Random sampling over joins revisited. In Gautam Das, Christopher M. Jermaine, and Philip A. Bernstein, editors, *Proceedings of the 2018 International Conference on Management of Data, SIGMOD Conference 2018, Houston, TX, USA, June 10-15, 2018*, pages 1525–1539. ACM, 2018.
