# OpenReview forum: "Sublinear Time Algorithms for Greedy Selection in High Dimensions"
_auai.org/UAI/2022/Conference — UAI 2022 Poster_

### Official Review · Reviewer_d8fE · 2022-04-13

**Q2(1) Originality/Novelty:** 3
**Q2(2) Significance/Impact:** 3
**Q2(3) Correctness/Technical Quality:** 3
**Q2(6) Clarity Of Writing:** 3
**Q6 Overall Score:** 6
**Q8 Confidence In Your Score:** 1

**Q1 Summary And Contributions:**

The authors propose sublinear time algorithms for greedy selections for high dimensional k-centre clustering and convex hull approximation.
There are theoretical guarantees of correctness of proposed methods.

**Q2 Assessment Of The Paper:**

More detailed information regarding each of these aspects is given below:

**Q2(4) Quality Of Experiments (Optional):**

3: Good: The experimental evaluation is adequate, and the results convincingly support the main claims.

**Q2(5) Reproducibility:**

3: Good: Key resources (e.g., proofs, code, data) are available and key details (e.g., proofs, experimental setup) are sufficiently well-described for competent researchers to confidently reproduce the main results.

**Q3 Main Strengths:**

The author s proposed  efficient algorithm for important problems in ML

**Q4 Main Weakness:**

I'm not sure about novelty. The proposed method seems to be just a small modification of existing ones

**Q5 Detailed Comments To The Authors:**

The paper is well written

**Q7 Justification For Your Score:**

I'm not convinced about novelty, but if authors or other reviewers convince me I will change my grade.

**Q9 Complying With Reviewing Instructions:**

1: Yes.

---

### Official Review · Reviewer_xiW8 · 2022-04-16

**Q2(1) Originality/Novelty:** 2
**Q2(2) Significance/Impact:** 2
**Q2(3) Correctness/Technical Quality:** 2
**Q2(6) Clarity Of Writing:** 2
**Q6 Overall Score:** 5
**Q8 Confidence In Your Score:** 3

**Q1 Summary And Contributions:**

This paper focuses on a new sublinear time algorithm for greedy methods in ML. In particular, this paper looks at k-center clustering and convex hull approximation, which can both be solved using greedy methods.

The paper employs strategies such as randomization and greedy selection to achieve the speedup. In particular, the proposed improvement can influence the running time of algorithms used for dataset selection, compression, active learning and topic modeling.



**Q2 Assessment Of The Paper:**

More detailed information regarding each of these aspects is given below:

**Q2(4) Quality Of Experiments (Optional):**

1: Poor: The experimental evaluation is flawed or the results fail to adequately support the main claims.

**Q2(5) Reproducibility:**

2: Fair: Key resources (e.g., proofs, code, data) are unavailable but key details (e.g., proof sketches, experimental setup) are sufficiently well-described for an expert to confidently reproduce the main results.

**Q3 Main Strengths:**

The paper is well written and carefully positions the novelty and compares it with the relevant literature.

The theoretical contribution is solid and addresses two fundamental problems that can have a significant impact in the ML community.

The main novelty of the proposed method over existing ones for submodular maximization [Mirzasoleiman 2015] is the difference in objective functions and the ability to deal with unknown k.



**Q4 Main Weakness:**

The complexity of Gonzalez’s algorithm is O(kdn), Huang et al is O(k^2/epsilon), and the proposed algorithm is O(k^2/epsilon). The improvement over Huaung et al. [2018] assumes that the dimension is very high, but in practice k may be larger than the dimension d.
In many subset selection problems with k-center we tend to choose more than 5% data, and the proposed algorithm may not perform well on these tasks.

In the CIFAR-10 experiment shown in Fig 1, 3072-dimensional embeddings were used.  The radius considered in Fig 1 is really large. Are the 3072-dimensions embeddings normalized? In many data subset selection and active learning problems, the embedding from the last layer are used as features. In ImageNet or CIFAR the embeddings vary from 64 to 2048. What is the point of representing an entire CIFAR-100 dataset with just 15 to 20 points?

I would expect some results on the generalization performance based on the test error when we do subset selection on the MNIST dataset.



**Q5 Detailed Comments To The Authors:**

Please address my list of weaknesses.

**Q7 Justification For Your Score:**

My main concerns are with the experiments as written above

**Q9 Complying With Reviewing Instructions:**

1: Yes.

---

### Official Review · Reviewer_tMBF · 2022-04-16

**Q2(1) Originality/Novelty:** 2
**Q2(2) Significance/Impact:** 2
**Q2(3) Correctness/Technical Quality:** 3
**Q2(6) Clarity Of Writing:** 4
**Q6 Overall Score:** 6
**Q8 Confidence In Your Score:** 3

**Q1 Summary And Contributions:**

In this paper, the greedy selection problems including k-center clustering and convex hull approximation are studied. The algorithms using randomized sampling are proposed, which guarantee to have bounded approximation ratios with high probability. The biggest advantage of the proposed algorithms is that their time complexities are independent of the sample size. Experiments are also conducted to verify their effectiveness.

**Q2 Assessment Of The Paper:**

More detailed information regarding each of these aspects is given below:

**Q2(4) Quality Of Experiments (Optional):**

3: Good: The experimental evaluation is adequate, and the results convincingly support the main claims.

**Q2(5) Reproducibility:**

4: Excellent: Key resources (e.g., proofs, code, data) are available and key details (e.g., proof sketches, experimental setup) are comprehensively described for competent researchers to confidently and easily reproduce the main results.

**Q3 Main Strengths:**

1. The paper is well-motivated. Removing the dependence of sample size for greedy selection is very meaningful for application in large-scale data analysis.

2. The paper is mostly technical sound. I have gone through most of the proofs, and find the results convincing.




**Q4 Main Weakness:**

1. The proposed algorithms actually achieve weaker versions of the approximation ratio than previous studies.



**Q5 Detailed Comments To The Authors:**

1. My biggest concern is about the relaxation of the definition of the approximation ratio. In previous studies such as [Huang et. al., 2018], the approximation ratio is defined under the condition that all instances in the sample are covered. While according to Theorem 1-3 in the paper, this condition is relaxed, and a parameter $\epsilon$ is introduced to control the number of possible instances that may not be covered (coverage rate). The current lacks clear clarification of this point since it is missing in the definition in Section 2. I think this is an unignorable difference since I think building sublinear time algorithms for guaranteeing that all instances are covered is much harder. Furthermore, the coverage rate is $(1-\epsilon)n$, which means that the number of instances that are not covered grows with the sample size. This may not be desired in practice. However, using the technique proposed in the paper, I think making the coverage rate independent of $n$ would make the dependence of time complexity to $n$ unavoidable.

2. In the proof of Theorem 2, I haven't understood why $i_{iter} \leq \tilde k$. In the paper, it said that "Since we relax the error of covering number to be $3ε > ε$, we know that  $i_{iter}$ should be no larger than $\tilde k$." But I think this inequality does not guarantee that at the phase that the stopping condition reaches, $i_{iter}$ could already exceed $\tilde k$, since it is possible to have that $\|P\backslash(\cup_{l=1}^i\mathbb B(c_l, r_0))\| < \epsilon n$.  To my understanding, we can only guarantee that $i_{iter} \leq 2\tilde k$, which is also another relaxation to the approximation ratio.

**Q7 Justification For Your Score:**

Overall, I think the paper is well-motivated and mostly technical sound. While I also think the relaxation of the definition to the approximation ratio is an unignorable drawback.

**Q9 Complying With Reviewing Instructions:**

1: Yes.

---

### Decision · Program_Chairs · 2022-05-15

**Decision:**

Accept (Poster)

**Comment:**

Meta Review: The paper offers a new method for greedy selection, applied to k-center clustering and convex hull approximation, with a runtime complexity which is independent of sample size. Reviewers found the contribution novel, important, and sound. The method's advantage lies mainly in the small-k regime, but this is an interesting enough regime and therefore this is not a fundamental drawback.
Some issues regarding the clarity of presentation came up during the review and discussion with the authors - I trust these will be amended by the authors in the final version accepted for publication.